# Expressive Flexibility and Mental Health: The Mediating Role of Social Support and Gender Differences

**DOI:** 10.3390/ijerph19010456

**Published:** 2022-01-01

**Authors:** Chenyu Shangguan, Lihui Zhang, Yali Wang, Wei Wang, Meixian Shan, Feng Liu

**Affiliations:** 1College of Education Science and Technology, Nanjing University of Posts and Telecommunications, Nanjing 210023, China; shanmx@njupt.edu.cn (M.S.); liuf@njupt.edu.cn (F.L.); 2Department of Psychology, Shanghai Normal University, Shanghai 200234, China; zhanglihui_psy@163.com; 3Department of Psychology, School of Marxist, Zhejiang University of Finance and Economics, Hangzhou 310018, China; yali_wang0513@zufe.edu.cn; 4School of Psychology, Central China Normal University, Wuhan 430079, China

**Keywords:** expressive flexibility, depression, life satisfaction, social support, gender differences

## Abstract

Recent research has emphasized the crucial role of expressive flexibility in mental health. This study extended prior studies by further exploring the mediating mechanism and possible gender differences underlying the association between expressive flexibility and mental health indexed by depression and life satisfaction based on the dual-factor model of mental health. Specifically, we explored whether social support mediated the association between expressive flexibility and depression as well as life satisfaction, and whether there were gender differences in these relationships. A total of 711 voluntary college students (mean age = 20.98 years, *SD* = 2.28; 55.70% women) completed a set of scales assessing expressive flexibility, perceived social support, depression, and life satisfaction. Results showed that expressive flexibility had a positive direct effect on life satisfaction and social support mediated this association. Social support also mediated the relationship between expressive flexibility and depression. The mediation effect of social support was robust and consistent in men and women whereas expressive flexibility had a stronger direct effect on depression in women compared to men. The present study contributes to clarifying the relationship between expressive flexibility and mental health from a more comprehensive perspective. Last, the strengths and limitations of this study were discussed.

## 1. Introduction

With the rise and development of positive psychology [1], a more comprehensive understanding of mental health has emerged among researchers and the general public. Both theoretical and empirical studies suggested that research on mental health should not be limited to the elimination of mental illness (e.g., depression), but should also focus on positive wellness (e.g., life satisfaction) [2,3]. Specifically, theorists in positive psychology have emphasized the importance of various positive traits and competencies for mental health [4]. As one of the most salient positive attributes of individuals, expressive flexibility has received increasing attention in recent years [5,6].

Expressive flexibility refers to the ability to flexibly enhance and suppress emotional expression in accordance with different contexts [7]. Previous studies have focused on investigating the relationship between expressive flexibility and negative aspects of mental health (e.g., depression, emotion distress, and psychology symptoms) [5,8,9,10], noting that expressive flexibility is a crucial protective factor for mental health. However, few examined the possible positive link between expressive flexibility and positive aspects of mental health, such as life satisfaction. Furthermore, what is unclear is the mechanism underlying the constructive impact of expressive flexibility on mental health and its boundary conditions. Therefore, based on the dual-factor model of mental health [2,11] and the social interaction model of emotion regulation [12], the present study extends previous research by including both depression and life satisfaction as outcome variables and social support as a possible mediating mechanism for the impact of expressive flexibility on mental health. Meanwhile, we also examined gender differences in these relationships. 

### 1.1. Expressive Flexibility 

In real life, individuals often need to respond flexibly to the ever-changing surrounding environment. Resilience is regarded as one key ability to successfully cope with these changes and adversities [13,14]. In this context, the concept of expressive flexibility was proposed to understand one’s ability to enhance and suppress emotional expressions to adapt to contextual demands [5,15,16]. Studies showed that resilience was positively associated with expressive flexibility [17], emotional flexibility [18], and coping flexibility [13]. Some researchers even identified expressive flexibility as one key component of resilience, highlighting the importance of flexibility in the regulation of emotions [19,20,21,22].

A vital theoretical framework for us to understand the process of emotional flexibility in human beings is the model of socioemotional flexibility at three times scale (the Flex3 model), which claims that socioemotional flexibility can be understood in terms of three distinct levels: micro, meso, and macro, corresponding to dynamic flexibility, reactive flexibility, and trait or developmental flexibility, respectively [23,24]. Based on this model, expressive flexibility ability can be regarded as socioemotional flexibility at the trait level and is beneficial to many desirable outcomes, such as learning, social competence, and creativity [24]. Thus, as one of the most salient positive attributes of individuals, expressive flexibility has attracted much attention and is worthy of further study. Based on the literature, expressive flexibility contains two crucial regulatory abilities: enhancement and suppression abilities [25]. Performance on either dimension alone is not representative of a person’s overall expressive flexibility ability. For example, people cannot be said to have a strong expressive flexibility ability if they are only able to enhance emotional expressions in response to the context but unable to suppress them. Accordingly, the present study approaches expressive flexibility as a holistic construct to reveal an individual’s ability to flexibly enhance and suppress one’s expressions, with the aim of further exploring the relationship between expressive flexibility and mental health (i.e., depression and life satisfaction).

### 1.2. Expressive Flexibility and Mental Health

Existing research has shown a clear relationship between expressive flexibility and mental health outcomes, especially in terms of psychopathology. For example, there is solid evidence that expressive flexibility deficits can predict mental illness, such as depression [9,10,26], emotion distress [8], and anxiety [27,28]. A recent survey among combat-exposed veterans suggested that lower levels of expressive flexibility were associated with more severe post-traumatic stress disorder (PTSD) and depression [9]. In comparison, however, there is far less research examining the impact of expressive flexibility on positive aspects of mental health, although there are some exceptions. For instance, a recent study found a positive relation between expressive flexibility and health-related quality of life in a Western elderly sample [29].

The dual-factor model of mental health (DFM) proposes that mental health should be a complete state, in which positive indicators such as subjective well-being should be considered in addition to traditional negative indicators such as mental illness [2,11]. Specifically, the applicability of life satisfaction to the wellness dimension and of depression to the illness dimension of DFM has been tested and the integrated indicators have been considered stable predictions of mental health in Chinese samples [30,31]. To our knowledge, only one study simultaneously examined the relation between expressive flexibility and mental health with both the positive and negative indicators in one sample. Specifically, Chen et al. found that higher expressive flexibility predicted fewer symptoms of depression and higher life satisfaction [6]. Thus, more evidence regarding the relationship between expressive flexibility and mental health with both negative and positive indicators should be obtained. In addition, most previous studies exploring the relationship between expressive flexibility and mental health have utilized Western samples. However, cultural differences in emotional expression and inhibition have been acknowledged in this field [32,33,34]. Although some investigations have provided positive evidence between expressive flexibility and mental health in Eastern samples [6,26], more evidence is needed to support the generalizability of the results.

Therefore, under the DFM framework, the primary aim of this study was to examine the relationship between expressive flexibility and two indicators of mental health (i.e., depression and life satisfaction), and to further enrich the cultural diversity of this research area by using a sample of Chinese university students in an Eastern cultural context. We hypothesize:

**Hypothesis** **1a.***Higher ability of expressive flexibility will be related to lower levels of depression*.

**Hypothesis** **1b.***Higher ability of expressive flexibility will be related to higher levels of life satisfaction*.

### 1.3. Mediating Role of Social Support

Although a series of studies explored the associations between expressive flexibility and relevant indicators of mental health, the mechanism underlying these relationships remains unclear. As mentioned earlier, expressive flexibility is a vital ability in social interaction contexts. As the social interaction model of emotion regulation assumes, the displaying of emotions in social contexts would exert reciprocal feedback [12]. Specifically, senders’ enhancement or suppression of emotions would influence receivers’ emotions and actions, which would in turn affect senders’ experience and actions. When the senders are sending emotions in accordance with the receivers or situations, this process might help improve the relationship between individuals and others, promote interpersonal communication, and increase perceived social support. In other words, expressive flexibility would be beneficial for senders, by helping them achieve interpersonal cooperation, and thus perceive and obtain more social support. 

The potential positive association between expressive flexibility and social support also coincides with both the social functions of emotional expressions and the theory of emotional intelligence. First, expressive flexibility involves an individual’s control and regulation of expressions, reflecting his or her socioemotional flexibility and playing a vital role in social contexts by transmitting information and communicating thoughts [35]. The degree to which emotional expression is enhanced or suppressed can regulate one’s social life [36], including the perceived social support. Second, emotional intelligence theorists believe that the ability to enhance and suppress emotional expressions in response to social situations (i.e., expressive flexibility) reflects a person’s social skills or socialization [37] and is an important aspect of emotional intelligence [38]. Many empirical studies have shown that higher emotional intelligence contributed to more social support [39,40], and higher socialization was also positively associated with more social support [41]. In this regard, both theoretical and empirical research suggested that expressive flexibility may be positively related to social support. However, to the best of our knowledge, no empirical studies provided direct evidence of the relationship between expressive flexibility and social support. 

Besides, abundant research has shown that social support could be a protective factor for depressive symptoms [42,43], and people who perceive more social support usually have higher life satisfaction [40,44]. Indeed, prior research has indicated that perceived social support mediated the relationship between emotional intelligence and mental health [40]. Following this line, perceived social support could also be a mediator in the relationship between expressive flexibility and mental health. Thus, we propose:

**Hypothesis** **2a.***Social support will mediate the relationship between expressive flexibility and depression*.

**Hypothesis** **2b.***Social support will mediate the relationship between expressive flexibility and life satisfaction*.

### 1.4. Gender Differences

Previous studies have indicated significant gender differences in expressive flexibility, perceived social support, and mental health (especially depression). Specifically, women are usually better at expressing and enhancing their emotions, whereas men prefer to suppress their emotions [45]. For example, a recent study on emotion regulation flexibility revealed gender differences in which women showed more flexibility in regulating their emotions compared to men [46]. In terms of perceived social support, a meta-analytic review of a total sample of over 3000 Chinese people indicated that women did perceive more social support than men [47]. Further, women with lower levels of social support were more likely to report more symptoms of depression and lower life satisfaction than men [48,49]. Although these gender differences have been explored, no studies have provided conclusive evidence of the potential gender differences in both the direct and indirect effects of expressive flexibility on mental health. Theoretically, the “tend-and-befriend” theory proposes that women are more flexible in expressing emotions in response to stress, as well as seeking and receiving more social support than men [50], which could protect women from depressive risks and contribute to psychological well-being [51]. In this regard, both the mediating effect of social support and the direct effect of expressive flexibility on mental would be greater for women than for men. Thus, we propose:

**Hypothesis** **3a.***The direct relationship between expressive flexibility and mental health (i.e., depression and life satisfaction) will be weaker for men than women*.

**Hypothesis** **3b.***The relationship between social support and mental health (i.e., depression and life satisfaction), as well as the indirect relationship between expressive flexibility and mental health mediated by social support will be weaker for men than women*.

### 1.5. The Present Study

To further our understanding of the effects of expressive flexibility on mental health and its underlying mechanism and boundary conditions, the present study constructed a moderated mediation model. In the model, expressive flexibility had opposing effects on positive and negative dimensions of mental health, i.e., depression and life satisfaction, which were mediated through social support, and these direct and indirect effects differed across men and women (see Figure 1). We subsequently tested this in a sample of Chinese university students.

## 2. Methods

### 2.1. Participants and Procedure 

The sample of the present study included 711 voluntary college students ranging from 17 to 33 years old (*M* = 20.98, *SD* = 2.28; 55.70% women) recruited both online and offline in China. The study protocol was approved by the Ethics Committee of the School of Psychology at ****, which was conducted in accordance with the code of ethics set by the Declaration of Helsinki and its later amendments. All the participants provided informed consent and received a brief introduction at the beginning of the survey. Then, they were required to complete a set of Chinese version questionnaires, including the Flexible Regulation of Emotional Expression Scale (FREE), Perceived Social Support Scale (PSSS), Beck Depression Inventory (BDI) and Satisfaction with Life Scale (SWLS). After the survey, they got monetary compensation (5 CNY) in gratitude for their participation. 

### 2.2. Measures

#### 2.2.1. Expressive Flexibility

Expressive flexibility was measured by the Chinese version of the Flexible Regulation of Emotional Expression Scale (FREE) with 16 items loaded onto two factors: enhancement with 8 items (e.g., “You receive a gift from a family member but it’s a shirt you dislike”) and suppression with 8 items (e.g., “You have just heard about the death of a close relative right before an important work meeting”) [6,25]. Participants were asked to rate how well they were able to express or conceal their emotions compared to their usual feelings in different contexts on a 6-point Likert scale ranging from 1 (unable) to 6 (very able). The expressive flexibility score was calculated by the following formula: (enhancement + suppression) − | enhancement–suppression | [6,25]. In this formula, the score of enhancement and suppression was separately calculated by summing up the scores of 8 items in each factor. In the present study, the coefficient alpha of the overall scale was 0.84, with α = 0.74 for the enhancement subscale, and α = 0.79 for the suppression subscale.

#### 2.2.2. Social Support

Social support was measured by the Chinese version of the Perceived Social Support Scale (PSSS) which contained 12 items [52,53]. The participants were asked to rate how they felt about each statement (e.g., “There is a special person with whom I can share the joys and sorrows”) from 1 (totally disagree) to 7 (totally agree). The score of PSSS was calculated by summing up all the scores of 12 responses. Internal consistency in this study was good (α = 0.95).

#### 2.2.3. Depression

Depression was assessed using the Chinese version of the Beck Depression Inventory (BDI) which contained 21 items [54,55]. Participants were required to choose one of the four statements about depression symptoms, which were then scored from 0 to 3. The final score of depression was calculated by summing up the scores of all items, ranging from 0 to 63. Higher scores indicated more severe symptoms of depression. The coefficient alpha in the present sample was 0.93.

#### 2.2.4. Life Satisfaction

Life satisfaction was measured by the Satisfaction with Life Scale (SWLS; Chinese version) [56,57] which was a 7-point Likert scale containing 5 items to measure global life satisfaction. Participants rated their agreement on five statements (e.g., “If I could live my life over, I would change almost nothing”) on a 7-point scale (1 = totally disagree, 7 = totally agree). Final scores were the sum of all items, ranging from 5 to 35, with higher scores indicating higher life satisfaction. The coefficient alpha in the present sample was 0.87.

### 2.3. Data Analysis

Data of the present study were analyzed with both SPSS 25 (IBM, Armonk, NY, USA) and Mplus 7.4 (Muthén & Muthén, Los Angeles, CA, USA). First, we conducted a common method bias test, descriptive analyses, and Pearson correlation analyses using SPSS 25. Second, we performed structural equation modeling to test the mediation effect of social support in the relationship between expressive flexibility and mental health (i.e., depression and life satisfaction) using Mplus 7.4 with the bias-corrected and percentile bootstrap based on 5000 samples [58,59]. If the value zero was not in the 95% confidence interval (CI), it indicated that the result was significant and the mediation model was supported. During this step, some demographic variables like age and grade were set as control variables [6,32]. Third, we conducted bias-corrected bootstrap tests and a Wald test using Mplus 7.4 to test the gender differences in both the indirect effect and the direct effect. All the data presented were standardized.

## 3. Results

### 3.1. Common Method Deviation Test

To test whether there was common methodological bias in the present study, the Harman single-factor test was applied [60]. The results showed that the explanatory variation of the first factor was 26.34%, which was lower than the maximum critical value 40%, indicating no obvious common methodological bias.

### 3.2. Descriptive Statistics and Correlations

Descriptive statistics and zero-order correlations among all variables were presented in Table 1. Expressive flexibility was positively associated with social support and life satisfaction, and negatively related to depression, which provided the initial evidence to support Hypotheses 1a and 1b. Moreover, social support was negatively associated with depression and positively related to life satisfaction, indicating mediation analyses could be performed.

### 3.3. Mediation Analyses

The results of mediation analyses (controlling for gender and age) were summarized in Figure 2. Expressive flexibility was positively and directly linked to life satisfaction and social support. Social support was negatively associated with depression and positively correlated with life satisfaction.

The bootstrapping method was then applied, and the results showed that social support mediated the relationship between expressive flexibility and depression, with the indirect effect = −0.08, SE = 0.01, 95% CI = [−0.12, −0.04], *p* < 0.001. The direct effect of expressive flexibility on depression was not significant, with the direct effect = −0.06, SE = 0.03, 95% CI = [−0.10, 0.01], *p* > 0.05. For life satisfaction, the results revealed that social support mediated the relationship between expressive flexibility and life satisfaction, with the indirect effect = 0.10, SE = 0.03, 95% CI = [0.05, 0.15], *p* < 0.001, indicating that greater expressive flexibility was related to more social support, and then related to higher life satisfaction. A significant direct effect of expressive flexibility on life satisfaction was also noted, with the direct effect = 0.15, SE = 0.03, 95% CI = [0.09, 0.20], *p* < 0.001.

In sum, the results showed that expressive flexibility predicted depression via social support. Expressive flexibility positively predicted life satisfaction directly, as well as indirectly through social support. Thus, Hypotheses 1a, 1b, 2a and 2b were supported.

### 3.4. Gender Differences

First, measurement equivalence (ME) analyses were conducted to confirm whether the constructs were measured in both the male and female groups in the same way. The results were summarized in Table 2. For all measures, the results showed that the difference fit indexes ΔCFI and ΔTLI for each model comparison were less than 0.01, satisfying scalar invariance and suggesting that all the measures for the constructs in the present study were equivalent across genders. For example, for life satisfaction, the change in CFI value of M3 compared to M2 was less than 0.01. After that, to test whether the mediating effect was consistent across genders, we tested the hypothesized model in women and men separately. The standardized coefficients were summarized in Figure 3. 

Based on the analyses above, bias-corrected bootstrap analyses were then conducted separately for men and women (see Table 3). The results showed that for both men and women, social support consistently mediated the relationship between expressive flexibility and depression as well as between expressive flexibility and life satisfaction (95% CIs did not include 0). For the direct effect, the effect of expressive flexibility on depression was significant in women but not significant in men.

In addition, to further test the consistency in the indirect path between expressive flexibility and mental health (with depression and life satisfaction as indicators) in men and women, the multi-group bias-corrected bootstrap test (controlling for age) was applied. The results showed no significant difference in the indirect effect between men and women in the relationship between expressive flexibility and depression (95% CI [−0.02, 0.09]), as well as that between expressive flexibility and life satisfaction (95% CI [−0.08, 0.01]), suggesting that the mediating role of social support between expressive flexibility and mental health was consistent in both men and women. 

Further, to test whether there were gender differences in the direct effect between expressive flexibility and mental health (i.e., depression and life satisfaction), a Wald test for equality constraints (both controlling for age) was applied. The results of the Wald test showed that there was a significant gender difference in the direct effect of expressive flexibility on depression, Wald test (1) = 8.36, *p* < 0.01; the gender difference in the direct effect of expressive flexibility on life satisfaction was not significant, Wald test (1) = 0.08, *p* > 0.05. The results revealed that greater expressive flexibility could directly reduce symptoms of depression in women, whereas such an effect has not been shown in men. In other words, gender moderated the relationship between expressive flexibility and depression. For clarification, we plotted the effect of expressive flexibility on depression at high (1 SD above the mean) and low (1 SD below the mean) levels of expressive flexibility separately (see Figure 4).

In sum, the results indicated that for the indirect effect of expressive flexibility on mental health (i.e., depression and life satisfaction), no gender differences were found. Regarding the direct effect, the relationship between expressive flexibility and depression in women was significantly greater than that in men. Thus, Hypothesis 3a was partly supported and Hypothesis 3b was not supported. 

## 4. Discussion

Previous studies have provided insufficient evidence regarding the relationship between expressive flexibility and mental health. The present study extended existing research by providing direct evidence for the relationship between expressive flexibility and mental health with both negative and positive indicators (i.e., depression and life satisfaction), testing the possible mediating role of social support and examining whether gender differences existed in these relationships in a sample of Chinese university students.

First, the present study supports the relationship between expressive flexibility and mental health. Specifically, our findings showed that expressive flexibility was negatively related to depression and positively associated with life satisfaction. These results are consistent with findings of previous studies [6,32], indicating that expressive flexibility could be a protective factor for psychological illness and a promoting factor for psychological well-being [26,61]. In other words, individuals with a higher ability of expressive flexibility are more likely to perceive higher life satisfaction and less likely to trigger a depressive risk. The findings also support the theory on socioemotional flexibility [23,24], which suggests that flexibility in social emotion will be beneficial to adaptive social well-being and mental health. 

Second, to further explore the mechanism underlying the relationship between expressive flexibility and mental health, we examined the mediating role of social support. The results showed that social support fully mediated the relationship between expressive flexibility and depression, and partially mediated the relationship between expressive flexibility and life satisfaction, which were not fully investigated by previous research to our knowledge. Thus, high perceived social support is one of the reasons why individuals with a greater ability of expressive flexibility have fewer depressive symptoms and are more likely to report higher life satisfaction. It is in accordance with previous studies reporting the mediating role of social support in the relationship between emotional intelligence and mental health in Chinese samples [40]. For the results regarding depression, the influence of expressive flexibility on depression was totally mediated through social support and no direct effect of mental health on depression was found, which is not fully consistent with previous studies [26,61]. One possible reason may be that there are additional moderating factors (e.g., gender) in the relationship between expressive flexibility and depression. Previous studies exploring the relationship between expressive flexibility and depression usually included gender as a control variable [6,61], thus the possible gender differences in this relationship were most likely to be concealed. Another possible reason may be that this study approached expressive flexibility as a whole psychological construct, whereas some previous studies only examined one dimension of expressive flexibility. For example, some found that lower levels of enhancement ability were related to more symptoms of depression [9]. 

Overall, our mediation analyses results suggest that flexible enhancement and suppression of emotional expressions in complex social situations affect different aspects of mental health by influencing the individuals’ perceived social support. The findings are consistent with the theoretical explanation of the social interaction model of emotion regulation [12]. From this view, expressive flexibility is one aspect of emotional capability that illustrates the importance of the enhancement and suppression of emotional expressions in social interactions [7], thus individuals with higher levels of expressive flexibility are more adaptable in social or interpersonal interactions [21], which in turn induces better social outcomes, especially higher perceived social support. In addition, many studies provided empirical evidence that expressive flexibility could predict adjustment in social interactions [5,7]. Moreover, based on the theory of the social function of emotional expression [35,62], emotional expressions conveying social information may affect others’ cognition and behaviors, and thus influence individuals’ social interaction as well as perceived social feelings. A recent meta-analysis on the expression and suppression of emotions also implied that proper expressions of emotions may lead to better social consequences and better perceived social cognition [63].

Third, our findings regarding gender differences showed that gender moderated the relationship between expressive flexibility and depression. Specifically, expressive flexibility had a negative effect on depression in women, whereas no such effect was found in men. That is, women with a higher ability of expressive flexibility are less likely to sink into depression. This can be explained by gender role expectations [64], which claim that women are expected to be more relationship-oriented than men, thus women may be more flexible to enhance and suppress emotional expressions to maintain good interpersonal relationships. In addition, it was found that good interpersonal relationships had a better buffering effect on depression in women than men, and women tended to be more psychologically beneficial from social relationships [65]. Therefore, combining the two aspects reveals higher levels of expressive flexibility help reduce depression in women. In contrast, males are seen as emotionally stable, tough, and less likely to get excited [66], which can be seen as more self-oriented. Protective factors against depression in men may often be self-related, so expressive flexibility does not directly predict depression in men. Moreover, the “tend-and-befriend” theory assumes that in the face of interpersonal demands and pressures, women would often flexibly regulate their emotional displays to receive more social support, which in turn may protect them from possible depressive risks and lead to a better perceived well-being [50]. To our knowledge, few studies explored the gender differences in this relationship, thus this link may be one novel finding in this field. The result suggests that expressive flexibility may be an even more important ability for women, with high expressive flexibility helping them reduce depression.

In sum, the present study advances relevant research in this field in the following ways. First, our findings provide clear evidence for the positive effect of expressive flexibility on mental health in Eastern samples, which could be meaningful for wider generalization as well as for enriching the cultural diversity in this field. Second, our study extends previous research by revealing that social support is one mechanism that may account for the relationship between expressive flexibility and mental health, and this contributes to our understanding of how expressive flexibility impacts mental health. Third, our findings suggest that expressive flexibility helps to reduce symptoms of depression in women more than men, which deepens our understanding of who would be more beneficial regarding the effect of expressive flexibility on mental health. This study contributes to the gender-specific mechanism between emotional factors and mental health in an Eastern sample, which would be beneficial to informing prevention measures. As for clinical or practical implications, our study emphasizes the importance of expressive flexibility on mental health. Specifically, expressive flexibility is one catalyst for life satisfaction and one protective factor for depression in women. This protective effect is mediated by social support, which implies one path by which expressive flexibility influences mental health. In addition, for the gender differences in the effect of expressive flexibility on depression, the preventions and interventions for depression should pay more attention to the effect of expressive flexibility in women rather than men. 

Besides the strengths and implications, the limitations of the present study should also be noted. First, we only explored the relationship between expressive flexibility and mental health with two indicators (depression and life satisfaction). More dimensions of mental health (e.g., psychopathology indicators like positive emotions, and psychological well-being indicators like loneliness) should be investigated to reveal the comprehensive constructs of expressive flexibility and mental health. This can be beneficial for us in building a more precise and comprehensive model, thus deepening our understanding of the relationship between expressive flexibility and a human’s mental well-being. Second, only college students were included in the present study, which may limit the generalizability of the results. Future studies could test the relationship between expressive flexibility and mental health in populations of different age groups. Third, from the perspective of methodology, the cross-sectional design adopted in the present study was limited in determining causality [67], thus more longitudinal and experimental designs should be employed to examine the constructs in the future [68]. Last, self-reported measures were applied in the present study. Although there was no obvious common methodological bias shown in the present study, social desirability may be induced. For example, there may be self-evaluation bias in rating one’s ability. People may tend to overestimate their ability in some situations (e.g., the Dunning–Kruger effect) [69]. Therefore, reported data from different information sources (e.g., friends, family, etc.) as well as experimental tasks to assess expressive flexibility more objectively could be considered in future research.

## 5. Conclusions

In conclusion, the present study offers valuable insights into the knowledge of expressive flexibility and mental health with a sample of emerging adults in Eastern cultures. Our findings suggest that expressive flexibility is a protective factor for mental health. Further, the mediation analyses suggest that social support can be an explanatory factor for the reason why expressive flexibility is related to depression and life satisfaction, which enriches our understanding of one possible mechanism underlying the relationship between expressive flexibility and mental health. The moderation analyses indicate a stronger direct relationship between expressive flexibility and depression in women than men, which highlights the importance of considering the effect of expressive flexibility on mental health differently between men and women.

## Figures and Tables

**Figure 1 ijerph-19-00456-f001:**
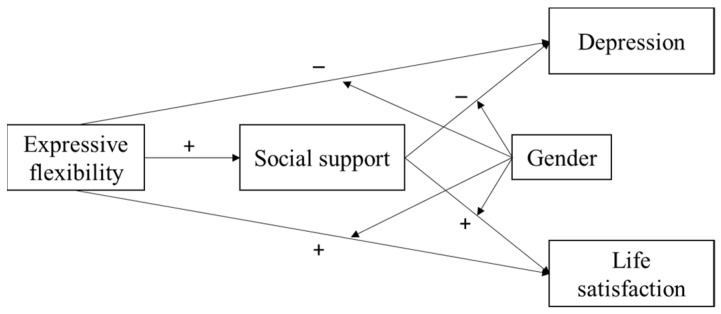
Hypothesized model.

**Figure 2 ijerph-19-00456-f002:**
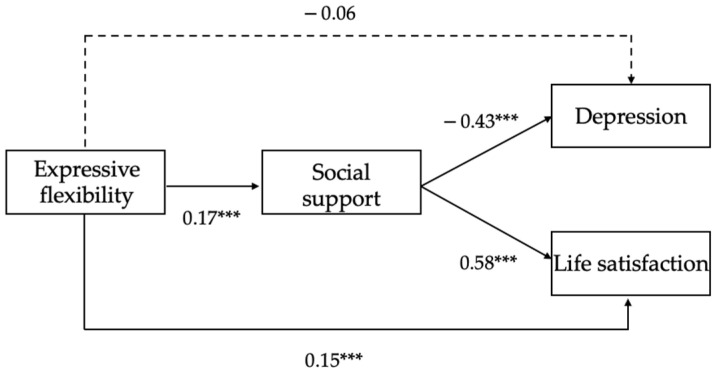
Mediation model of expressive flexibility, social support, depression, and life satisfaction (*N* = 711). Note: *** *p* < 0.001.

**Figure 3 ijerph-19-00456-f003:**
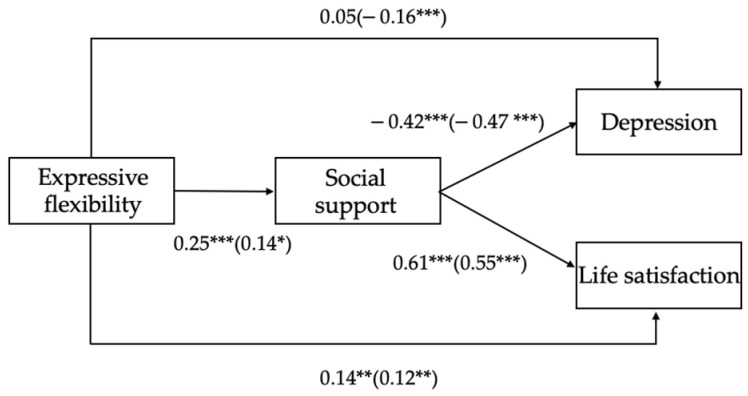
Mediation model between men (*N* = 315, coefficients outside the brackets) and women (*N* = 396, coefficients inside the brackets). Note: * *p* < 0.05, ** *p* < 0.01, *** *p* < 0.001.

**Figure 4 ijerph-19-00456-f004:**
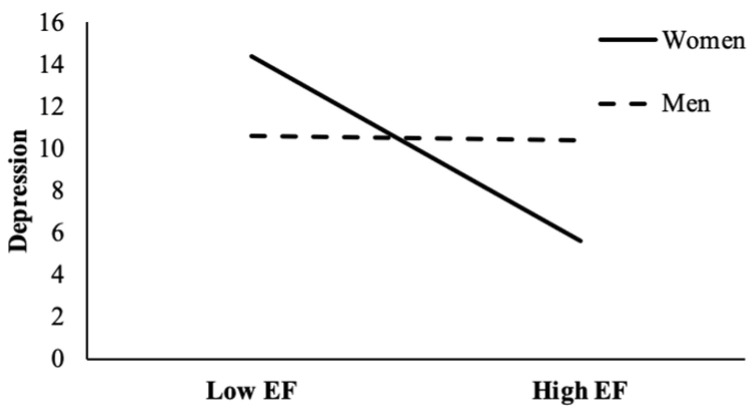
Gender moderated the relationship between expressive flexibility and depression. EF = expressive flexibility.

**Table 1 ijerph-19-00456-t001:** Means, standard deviations, and correlations among the variables in the study (*N* = 711).

Variables	*M* (*SD*)	1	2	3	4
1. Expressive Flexibility	58.32(13.62)	(0.84)			
2. Social Support	60.09(14.12)	0.18 **	(0.95)		
3. Depression	11.34(10.17)	–0.13 **	–0.44 **	(0.93)	
4. Life Satisfaction	21.88(6.22)	0.25 **	0.59 **	–0.48 **	(0.87)

Note: Cronbach’s α reliability coefficients are presented in parentheses on the diagonal. ** *p <* 0.01.

**Table 2 ijerph-19-00456-t002:** Results of measurement equivalence of all scales between genders.

Model	χ^2^*/df*	CFI	TLI	RMSEA	SRMR
Expressive Flexibility					
M1 Configural Invariance	2.47	0.91	0.89	0.06	0.06
M2 Metric Invariance	2.39	0.91	0.89	0.06	0.06
M3 Scalar Invariance	2.46	0.90	0.89	0.06	0.06
Social Support					
M1 Configural Invariance	3.85	0.96	0.95	0.09	0.04
M2 Metric Invariance	3.55	0.96	0.95	0.09	0.04
M3 Scalar Invariance	3.43	0.96	0.95	0.08	0.04
Depression					
M1 Configural Invariance	2.53	0.91	0.90	0.07	0.05
M2 Metric Invariance	2.50	0.91	0.90	0.07	0.06
M3 Scalar Invariance	2.48	0.90	0.90	0.07	0.06
Life Satisfaction					
M1 Configural Invariance	1.22	0.99	0.99	0.05	0.01
M2 Metric Invariance	1.55	0.99	0.99	0.04	0.03
M3 Scalar Invariance	2.00	0.99	0.99	0.05	0.02

**Table 3 ijerph-19-00456-t003:** Mediation results of bias-corrected bootstrap analyses in men and women (controlling age).

Path		Indirect Effect (SE)	Direct Effect (SE)	95% CI for Indirect Effect
Path_ED	Men	−0.10 ***(0.03)	0.05(0.05)	[−0.17, −0.05]
	Women	−0.06 *(0.03)	−0.16 ***(0.04)	[−0.12, −0.01]
Path_EL	Men	0.15 ***(0.04)	0.14 **(0.04)	[0.08, 0.23]
	Women	0.08 *(0.03)	0.12 **(0.03)	[0.02, 0.14]

Note: Path_ED: Expressive flexibility → social support → depression (social support as the mediating variable), Path_EL: Expressive flexibility → social support → life satisfaction, * *p* < 0.05, ** *p* < 0.01, *** *p* < 0.001.

## Data Availability

The data of the present study supporting the conclusions will be made available on request by the corresponding authors.

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
