# Peer review of "Expressive Flexibility and Mental Health: The Mediating Role of Social Support and Gender Differences"

_ijerph, 2022, doi:10.3390/ijerph19010456_

Round 1

Reviewer 1 Report

Review: Expressive Flexibility and Mental Health: The mediating role of Social Support and Gender Differences.

The study examined how positive wellness and traits contribute to life satisfaction. It employed a large sample from a relatively easy population and has a close mean split in gender participants, with relatively predictable results. What makes the study interesting, is the link between expressive flexibility and poor mental health. The study suggests that maintaining positive, flexible approach, may enhance/prevent mental illnesses, such as depression. The methodology is well explained and comprehensive enough, the results and conclusions are in line with the hypotheses and the authors highlight the limitations of the study.

Although the topic is safe, I found the approach pertinent and of interest and have the opinion that it has the potential to make a valuable contribution. It is a fitting topic and empowering in the times we live in.

  • Quality of Presentation: The review is well presented and of academic value.
  • Interest to the Readers: It is my opinion that the article of value to academic and lay readers.
  • Overall Merit: This study has merit in publishing.
  • English Level: Level of English is poor and needs to be edited.

References

  • References confirm to journal standard.
  • Good selection of current and older sources.

Strengths of the paper

  • The study can have an impact on the maintenance of good mental health
  • Constructed model of proposed moderated mediation.
  • Study made use of a wide variety of instruments.
  • The study is methodologically strong

Weakness of the paper

  • The manuscript makes use of many useless conjunctions that make reading difficult. Sometimes expressing using simple words get the message across easier. In academia the message is more important than making use of elaborate words.
  • The many different hypotheses make the study difficult to follow. When attempting to educate, it is advised to keep it simple for the reader to follow.
  • Although sample is large enough, population makes it difficult to generalize?

Major Points

  • For the sake of uninformed readers, I suggest defining “expressive flexibility” as articles can also serve as educating readers. (Line 40, p.1)
  • Line 70 to 71 refers to “two distinct substantial forms of emotional regulation abilities without mentioning it. I suggest being clear about this as it could be confused with the two sub-dimensions mentioned earlier.
  • Line 213: Please capitalize each word of Perceived Social Support Scale (as it is a name)

Minor Points

  • Line 53 – 55 on p2, is basically saying the same thing, please rephrase to be clear on the intent of the sentence.
  • Line 61 – 62, p2, is similarly unclear.
  • Perhaps line 231/2322 should read: “Previous studies have provided evidence for the relationship between expressive flexibility and mental health, but few indicated positive evidence”? The sentence is unclear.

Author Response

Thank you very much for your time and work.

Reviewer 2 Report

The study presented in this paper has the merit of focusing on the processual/ mediational role of social support in the relationship between expressive flexibility and mental health. Furthermore, it explores gender as a moderating variable in the afore mentioned relationship.

The study is well intoduced and contextualized within the existing literature.

It is also methodologically sound, although in this respect there might be room for improvement:

  1. It would be important to have the fit indices for the model presented in Fig. 2 (X2/df; GFI; CFI; RMSEA);
  2. When testing the gender invariance of the model, it would also be important to present the fit indices of the basal model as well as the invariance indices for the structutal weights, the structural covariances and structural residuals.

The discussion section is appropriately organized and the study's results are theoretcially and practically explored in a satisfactory manner.

Overall, the paper is scientifically sound and robust.

Yet, the presentation of the afore mentioned results (the path analysis model and invariance fit indices), would mean a significant improvement to the paper's quality.

Author Response

Thank you for your time and work.

Reviewer 3 Report

The authors presented interesting research confirming the importance of emotional flexibility for mental health. The demonstration of a link between the level of EF ability and depression and life satisfaction may provide a basis for designing further relevant prevention and interventions for depression. However, the paper also has some weaker points.
In the theoretical part, the authors should consider doing a bit of tidying up to avoid the repetitions that are present. It also seems that it may be worthwhile to present EF in the broader context, e.g. by mentioning its relation to resilience.
Hypotheses 1 a and 1 b seem to be formulated in a too simplified way, it seems that perhaps the formulation "higher ability of expressive flexibility" would be more precise than "expressive flexibility" alone.
A weakness of the survey is the sample selection. Student groups are not a representative example, moreover all generalizations that appear in the conclusions should refer to young people, as perhaps in later years of a person's life EF has a different meaning.
The authors should also read the text more carefully and correct sentences that are too vaguely written and, unfortunately, phrases that do not make any sense like this one: 
"we extended existing literature by showing that social support fully mediated the relationship between expressive flexibility and depression and partially mediated the relationships between expressive flexibility and depression" (432 and below)

Author Response

Thank you for your time and work.
